# iPSC-Derived Biological Pacemaker—From Bench to Bedside

**DOI:** 10.3390/cells13242045

**Published:** 2024-12-11

**Authors:** Quan Duy Vo, Kazufumi Nakamura, Yukihiro Saito, Toshihiro Iida, Masashi Yoshida, Naofumi Amioka, Satoshi Akagi, Toru Miyoshi, Shinsuke Yuasa

**Affiliations:** 1Department of Cardiovascular Medicine, Okayama University Graduate School of Medicine, Dentistry and Pharmaceutical Sciences, Okayama 700-8558, Japan; dr.duyquan@gmail.com (Q.D.V.); pqwy461x@s.okayama-u.ac.jp (T.I.); yoshid-m@cc.okayama-u.ac.jp (M.Y.); akagi-s@cc.okayama-u.ac.jp (S.A.); miyoshit@cc.okayama-u.ac.jp (T.M.); yuasa@okayama-u.ac.jp (S.Y.); 2Center for Advanced Heart Failure, Okayama University Hospital, Okayama 700-8558, Japan; 3Department of Cardiovascular Medicine, Okayama University Hospital, Okayama 700-8558, Japan; p5438a3l@s.okayama-u.ac.jp (Y.S.); pjjb912y@s.okayama-u.ac.jp (N.A.)

**Keywords:** sinoatrial node, HCN channels, induced pluripotent stem cell

## Abstract

Induced pluripotent stem cell (iPSC)-derived biological pacemakers have emerged as an alternative to traditional electronic pacemakers for managing cardiac arrhythmias. While effective, electronic pacemakers face challenges such as device failure, lead complications, and surgical risks, particularly in children. iPSC-derived pacemakers offer a promising solution by mimicking the sinoatrial node’s natural pacemaking function, providing a more physiological approach to rhythm control. These cells can differentiate into cardiomyocytes capable of autonomous electrical activity, integrating into heart tissue. However, challenges such as achieving cellular maturity, long-term functionality, and immune response remain significant barriers to clinical translation. Future research should focus on refining gene-editing techniques, optimizing differentiation, and developing scalable production processes to enhance the safety and effectiveness of these biological pacemakers. With further advancements, iPSC-derived pacemakers could offer a patient-specific, durable alternative for cardiac rhythm management. This review discusses key advancements in differentiation protocols and preclinical studies, demonstrating their potential in treating dysrhythmias.

## 1. Introduction

Arrhythmias, characterized by irregular or abnormal heart rhythms, are a significant contributor to morbidity and mortality worldwide [1]. Central to maintaining regular heart rhythm is the sinoatrial node (SAN), a specialized group of pacemaker cells responsible for initiating the heartbeat through spontaneous electrical impulses. Disruption in the function of the SAN can result in bradyarrhythmias. Atrioventricular (AV) block, another common arrhythmia, occurs when the electrical signal from the atria to the ventricles is partially or completely blocked, impairing synchronized cardiac contraction and reducing effective blood flow. These conditions often necessitate external interventions to stabilize rhythm. Traditionally, electronic pacemakers have been the gold standard for managing symptomatic arrhythmias such as atrioventricular block or SAN dysfunction [2]. In the United States alone, over 350,000 pacemaker procedures are conducted every year, with sinus node dysfunction accounting for more than half of these implantations [3].

Pacemakers are typically effective and reliable, benefiting from continuous advancements in technology. However, they are not without limitations. A primary concern is the potential for device malfunction, which can arise from issues such as loose or faulty leads, battery exhaustion, or interference caused by electromagnetic fields. The incidence of post-implantation complications is 9% in the first month and 6% over the following years [4]. Another complication is device infection, which, although rare, is a serious issue linked to significant morbidity and an elevated risk of mortality [5]. Furthermore, cardiac pacing in newborns and infants is particularly risky because their great vessels are often too narrow or inaccessible due to congenital malformations [6]. Additionally, they are associated with complications such as lead fractures and pacemaker-induced cardiomyopathy, especially with long-term use in younger patients [7]. These challenges underscore the need for alternative therapeutic strategies that can provide more natural and sustainable solutions for rhythm management [8].

The concept of a biological pacemaker has emerged as a promising alternative to electronic devices. Biological pacemakers aim to restore the heart’s rhythm using cells capable of spontaneous electrical activity, similar to the native SAN [9]. Induced pluripotent stem cells (iPSCs), which can be reprogrammed from adult somatic cells and differentiated into various cell types, have become a focal point in this endeavor [10,11]. Unlike gene-based approaches that modify existing heart cells and face limitations—such as host cell maturity, potential off-target effects, and the challenge of sustained rhythm control—iPSC-derived pacemaker cells offer distinct advantages. These cells can be generated in vitro and transplanted to form new pacemaker tissue, developing specialized cells that naturally exhibit pacemaker activity, independent of host cells, thus providing more stable and consistent rhythm regulation [12,13]. Preclinical studies have shown that iPSC-derived cardiomyocytes can successfully integrate into host cardiac tissue and restore rhythmic activity in models of heart block, demonstrating their potential as a biological alternative to electronic pacemakers [14,15] (Figure 1).

Despite these promising developments, significant challenges remain in the translation of iPSC-derived biological pacemakers to clinical use. Issues such as ensuring the purity of differentiated cells, managing cellular heterogeneity, and achieving stable long-term functionality must be addressed [16,17]. Furthermore, understanding the complex interplay of ion channels and cellular mechanisms that underlie spontaneous pacemaking is crucial for optimizing the performance of iPSC-derived pacemaker cells [18]. This review aims to explore the current state of iPSC-derived biological pacemakers, detailing the differentiation strategies, preclinical successes, and challenges that need to be overcome for their clinical application.

## 2. iPSC in the Realm of Arrhythmia

The discovery of iPSCs has marked a significant milestone in regenerative medicine, offering a versatile platform for studying disease mechanisms and developing innovative therapeutic approaches [19]. iPSCs are generated by reprogramming adult somatic cells, such as skin fibroblasts and peripheral blood cells, back to a pluripotent state through the introduction of specific transcription factors (OCT4, SOX2, KLF4, and MYC) [20]. This ability to generate patient-specific pluripotent cells holds considerable promise in the field of cardiovascular medicine, particularly in the development of biological pacemakers designed with the aim of treating cardiac arrhythmias [21,22,23].

The SAN generates electrical impulses that maintain a steady heart rhythm through a complex interplay between the “membrane clock”—comprising ion channels and transporters on the cell membrane—and the intracellular “Ca^2+^ clock”, which involves subcellular calcium-handling mechanisms [24,25]. When SAN function is compromised due to disease or aging, the heart’s ability to regulate its rhythm is impaired, leading to conditions such as bradycardia and sinus node dysfunction [26].

Traditional treatment for these conditions involves the use of electronic pacemakers, which deliver electrical impulses to regulate the heart rate. However, as discussed, these devices are not without their drawbacks. In contrast, iPSC-derived biological pacemakers offer a novel approach by utilizing stem cell technology to create functional pacemaker cells capable of autonomous electrical activity. iPSCs can be differentiated into various cardiac cell types, including nodal-like cells that mimic the properties of SAN and AV node cells, by guiding them through specific developmental pathways. This process often involves the modulation of signaling pathways such as FGF, retinoic acid, and nodal/activin [25,26,27]. These factors drive the differentiation of iPSCs into cardiac mesoderm, ultimately yielding pacemaker-like cells that exhibit spontaneous depolarization and rhythmic contraction.

## 3. Mechanisms of Pacemaking in iPSC-Derived Cardiomyocytes

SAN is distinct in its cellular composition, mechanical structure, and electrical activity when compared to other regions of the heart. Its pacemaker activity originates from nodal cardiomyocytes, which generate rhythmic electrical impulses that initiate myocardial contraction. SAN cardiomyocytes differ significantly from chamber cardiomyocytes. Due to the lower mechanical demands, pacemaker cells exhibit less-organized contractile units, fewer mitochondria, and typically present as spindle- or spider-shaped forms with a smaller cytoplasm. They also have a unique protein profile, such as low-conductance connexin (Cx) 45, in contrast to the higher-conductance Cx43 or Cx40 observed in other cardiomyocytes [28,29]. The expression of specific ion channels underlies two proposed mechanisms of automaticity: the calcium clock and the membrane clock. There is evidence suggesting that they have interacting roles [30]. The interplay between the If current and calcium cycling within the cell is often described as the “coupled-clock” system, where spontaneous local Ca^2+^, released from the SR, interacts with membrane ion channels to generate the rhythmic action potentials of pacemaker cells [31]. The synchronized function of various ion channels is crucial for the generation of stable pacemaker activity (Figure 2).

The ability of iPSC-derived cardiomyocytes to generate spontaneous electrical impulses is crucial for their potential application as biological pacemakers. This pacemaking activity is driven by intricate interactions between ion channels and intracellular signaling pathways, closely resembling the characteristics of native SAN cells. A deeper understanding of these mechanisms is essential for optimizing iPSC-derived pacemaker cells and ensuring their effectiveness in restoring normal heart rhythms.

### 3.1. The Membrane Clock Mechanism

The primary factors driving spontaneous pacemaking in both native SAN cells and iPSC-derived cardiomyocytes are a range of cell-surface ion channels, including hyperpolarization-activated cyclic-nucleotide-gated (HCN) channels, L-type and T-type voltage-gated calcium channels, and delayed rectifier potassium channels. HCN channels, especially HCN4 isoforms, are the most prevalent for the sarcolemma of pacemaker cells and play a critical role in initiating the diastolic depolarization that triggers spontaneous action potential firing [32,33]. As the cell membrane becomes more polarized, HCN channels increase their probability of opening, creating a slow inward current during the diastolic phase. This depolarizing current, known as the “funny” (I(f)) current, is activated as the membrane potential grows more negative, allowing sodium (Na^+^) and potassium (K^+^) ions to enter the cell, gradually moving the membrane potential closer to the action potential threshold. Following this, delayed rectifier potassium channels produce an outward current that brings the membrane potential back to a range where HCN channels can reopen, thus initiating a new cycle [34].

A significant limitation of using iPSC-derived cardiac cells as a model for studying working cardiomyocytes lies in their immature electrical properties [35]. In 2021, Federica Giannetti demonstrated that, under standard culture conditions, human iPSC-derived cardiomyocytes (hiPSC-CMs) predominantly expressed HCN4 and HCN1 isoforms. The density of the *I*_f_ current remained stable across different stages of cardiomyocyte differentiation (days 15, 30, and 60). However, after day 30, the voltage dependence of activation shifted to more negative potentials, and the activation time constants (τ) became slower. This shift contributed to a gradual decrease in the cells’ spontaneous beating rate over time [36]. Other research from Yukihiro Saito in 2022 illustrated that overexpressing HCN4 in hiPSC-CMs improved their pacemaker function by increasing the If current, resulting in higher spontaneous firing rates and more stable pacing activity [37]. These studies suggest that adjusting HCN channel expression in iPSC-derived pacemaker cells may be a promising approach to improve their capacity to replace the native SAN in cases of dysrhythmia.

### 3.2. The Calcium Clock Mechanism

In addition to HCN-mediated *I*_f_ currents, the “calcium clock” mechanism is essential for pacemaker cell automaticity. This mechanism relies on rhythmic calcium release from the sarcoplasmic reticulum (SR) through ryanodine receptors, causing transient increases in intracellular calcium levels [38]. This calcium release activates the sodium–calcium exchanger (NCX1), which expels one calcium ion in exchange for three sodium ions, thereby elevating the membrane potential to the threshold needed to open voltage-gated calcium channels and trigger depolarization [39]. Following this, the SERCA2 pump reloads calcium into the SR, readying the cell for the next cycle. The autonomic nervous system closely modulates the action potential rate by binding neurotransmitters to G-protein-coupled receptors (β-adrenergic and muscarinic receptors), activating pathways such as cAMP and PKA signaling to adjust heart rate [40].

In fact, the membrane clock and the calcium clock do not operate independently; instead, functional interactions between them are essential for normal pacemaker activity. Surface membrane proteins influence both membrane potential and intracellular Ca^2+^ cycling, while Ca^2+^ cycling proteins reciprocally affect the membrane potential through Ca^2+^-modulated surface electrogenic molecules [41]. Additionally, coupling factors such as PKA and CaMKII phosphorylation, which impact proteins in both clocks, play a crucial role in regulating this coupled-clock system for normal automaticity [42,43]. These unique SAN properties offer valuable insights for biological pacemaker designs.

### 3.3. Other Ion Channels in Pacemaking

In addition to HCN channels and calcium channels, sodium channels are crucial in shaping the action potentials of iPSC-derived pacemaker cells. Potassium currents, such as the inward-rectified K^+^ current, play a key role in regulating the resting membrane potential and ensuring appropriate action potential duration [44]. For iPSC-CMs, reduced expression of IK1 is advantageous as it promotes easier depolarization, supporting the spontaneous activity of these cells, similar to the native SAN [45]. Achieving a balance between depolarizing and repolarizing currents is critical for maintaining the rhythmic firing of pacemaker cells, ensuring their functionality when transplanted into the heart. However, hiPSC-CMs can sometimes express ion channels that are not typically present in cardiac cells. For example, big-conductance calcium-activated potassium currents (IBK, Ca) have been implicated in causing induced after-depolarizations and oscillations in engineered heart tissue (EHT) derived from an hiPSC line, C25 [46]. These occurrences can result from genetic alterations during the cell culture process. Kilpinen et al. highlighted that chromosome 10, which contains BKCa/KCNMA1, is one of the loci most prone to copy number alterations [47].

## 4. Differentiation Protocols for Making Pacemaker-like Cells

The ability to derive functional pacemaker cells from iPSCs hinges on efficient differentiation protocols that replicate the developmental pathways of SAN. These protocols aim to produce cells with the electrophysiological and molecular characteristics of native SAN pacemaker cells, making them suitable for use as biological pacemakers. Recent advancements in stem cell research have led to a variety of methods that promote the differentiation of iPSCs into pacemaker-like cells, yet challenges remain in achieving high efficiency, purity, and consistency.

### 4.1. SAN Development

The development of the SAN is a complex process in which pacemaker cells rapidly gain their distinctive characteristics during early embryonic stages, unlike the remaining myocardium, which continues to mature postnatally. Pacemaker cells originate from the lateral plate mesoderm, with fate mapping studies indicating that their lineage is determined shortly after gastrulation [48,49]. This specification process occurs before visible heart structures form and before classic cardiac markers are expressed. The early differentiation of pacemaker cells is influenced by signaling pathways, such as those of retinoic acid (RA), bone morphogenic proteins (BMPs), and Wnt [50,51,52]. During this time, pacemaker cells begin expressing unique transcription factors, such as ISL1, TBX18, TBX3, and SHOX2 [53,54,55], marking their divergence from other cardiac cells. As development continues, they integrate into the sinus venosus, the precursor to the right atrium, while expressing the ion channels and proteins necessary for their rhythmic activity [56].

The ability of pacemaker cells to generate and propagate action potentials depends not only on their inherent properties but also on overcoming the electrochemical constraints posed by their environment. These cells have a less negative resting membrane potential compared to atrial cells, creating challenges for impulse transmission [57]. The SAN must overcome a “source-sink” mismatch, where the small number of pacemaker cells must generate sufficient current to influence the larger atrial myocardium [58,59]. To address this, the SAN reduces electrical coupling through a lower expression of high-conductance gap junction proteins such as CX43 and CX40, while increasing the expression of low-conductance channels such as CX45, ensuring that pacemaker cells retain their charge longer and function autonomously [58].

As the SAN develops, it undergoes structural changes that further promote electrical insulation. After pacemaker cells differentiate, mesenchymal cells invade the region, forming a collagen-rich extracellular matrix (ECM) [56]. This remodeling creates clusters of pacemaker cells with reduced connectivity to the surrounding myocardium. The ECM and changes in cellular architecture help to preserve the pacemaker’s rhythm and prevent conduction block at the SAN–atrium junction [56]. Disruptions in this remodeling process can lead to irregular heart rhythms, emphasizing the importance of a well-patterned environment for pacemaker function [59].

### 4.2. Overview of Strategies to Develop Pacemaker-like Cells

The differentiation of iPSCs into cardiac pacemaker cells involves guiding the cells through stages that mimic embryonic heart development. This process typically begins with the induction of cardiac mesoderm, followed by further specialization into pacemaker-like cells. Key steps include the modulation of signaling pathways such as Wnt/β-catenin, BMPs, and transforming growth factor-beta (TGF-β), which are crucial for specifying cardiac progenitors [60]. To induce a pacemaker-like phenotype, differentiation protocols often incorporate factors that are known to influence SAN development.

One prominent approach is functional genetic re-engineering, which involves introducing pacemaker-related genes into quiescent ventricular cardiomyocytes to induce spontaneous electrical activity. This method aims to replicate the function of pacemaker cells by expressing genes such as HCN channels, β2-adrenergic receptors, and mutant KIR2.1 channels, either individually or in combination [61,62,63]. These genetic modifications can induce spontaneous action in non-pacing cells; however, the approach faces challenges such as arrhythmic complications and difficulty in achieving stable heart rates, which has led to the exploration of mutant-gene-based or dual-gene strategies [64,65,66].

Another strategy is the hybrid approach, where genes associated with pacemaker activity are introduced into non-cardiomyocyte cells, including human mesenchymal stem cells (hMSCs), human cardiomyocyte progenitor cells (hCPCs), or fibroblasts [67,68]. These engineered cells can be transplanted into cardiac tissue, where they interact with adjacent ventricular myocytes (VMs) through cell fusion or electrical coupling, influencing local cardiac activity [69]. The incorporation of HCNs (HCN1, 2, and 4) into non-cardiomyocyte cells has the potential to induce pacemaker ion currents (If) in in vitro cell models. This approach benefits from the immune-privileged properties of certain stem cells, but it requires time for the engineered cells to establish functional connections and exhibits limitations such as low basal heart rates and concerns about cell migration and differentiation [67,70].

A third approach involves the direct reprogramming of VMs into pacemaker-like cells using transcription factors such as TBX18 or TBX3, which mimic the natural development processes of SAN [71]. This method induces a comprehensive change in cellular structure and function, closely resembling genuine pacemaker cells. Initial studies have demonstrated successful automaticity in animal models, and advances in delivery methods, such as adeno-associated viruses and chemically modified mRNAs, have reduced immune responses and improved clinical applicability [72,73,74].

Recently, iPSCs have gained attention as a potential approach for creating biological pacemakers. These cells can be guided into becoming pacemaker-like cardiomyocytes through either transgene-dependent or transgene-free strategies. Transgene-dependent techniques involve genetic modifications, such as inducing MYC expression with doxycycline and inhibiting NODAL, which helps suppress NKX2-5 and enriches populations of SAN-like cells [75,76]. Another strategy involves modulating Wnt/β-catenin signaling to boost the expression of SAN-specific markers such as ISL1 and TBX18 [76]. In contrast, transgene-free approaches focus on altering signaling pathways without genetic alterations; they employ elements such as Wnt/β-catenin inhibitors BMP4, RA, and NODALor using co-culturing methods with mouse visceral endoderm-like cells (END-2) to drive SAN-like cell differentiation [12,27,77,78]. Moreover, the incorporation of cadherin-5 (CDH5) has been found to aid in the differentiation of pacemaker cells [78]. Each protocol seeks to recreate the developmental microenvironment of the sinoatrial node to generate hiPSC-derived pacemaker cells, with ongoing research aiming to refine these strategies for enhanced efficiency, purity, and functionality (Figure 3). Table 1 shows the advantages and limitations of each protocol.

### 4.3. Challenges in Inducing SAN-like Cells

One of the key challenges in creating iPSC-derived biological pacemakers is achieving a high concentration of pacemaker-like cells within the differentiated population [79]. Differentiation protocols often result in a heterogeneous mixture of atrial, ventricular, and pacemaker-like cells, which can complicate their application in clinical settings. Another difficulty lies in maturing iPSC-derived pacemaker cells to attain the functional characteristics of adult SAN cells. These cells often exhibit immature electrophysiological traits, such as reduced upstroke velocities and extended action potential durations [35].

To address this, strategies such as fluorescence-activated cell sorting (FACS) based on specific surface markers or genetic reporters linked to crucial pacemaker genes, such as HCN4, are employed [80]. Furthermore, gene-editing techniques such as CRISPR/Cas9 offer ways to improve the precision and effectiveness of differentiation protocols by targeting genes essential for SAN development, resulting in more consistent and functional pacemaker cells [81]. Furthermore, bioengineering methods, including 3D bioprinting and tissue engineering, can be used to create environments that closely resemble the natural cardiac niche, thereby supporting the maturation of iPSC-derived pacemaker cells.

## 5. Preclinical Studies of SAN-like Cell Treatment

The successful clinical translation of iPSC-derived pacemaker cells requires comprehensive preclinical testing to ensure their safety, efficacy, and proper integration with host cardiac tissues (Table 2). Animal models, including guinea pigs, dogs, and pigs, are essential for evaluating these cells, providing a bridge between laboratory research and therapeutic applications. One notable example is the work by Eduardo Marban’s team, which first introduced the concept of biological pacemakers using viral vectors to suppress IK1 in guinea pigs. This suppression enabled the modified myocytes to generate spontaneous, rhythmic electrical activity akin to natural pacemaker cells. In vivo testing revealed that, while some animals maintained normal sinus rhythms with prolonged QT intervals, others developed spontaneous ventricular rhythms, originating from the modified cells [64]. Despite the promise of this approach, it raised concerns about potential arrhythmic risks and variability in outcomes, highlighting the importance of controlling gene expression and ensuring stable integration with host tissue.

In another approach, Protze et al. developed SAN-like pacemaker cells (SANLPCs) through a transgene-independent protocol that involved the use of BMP and RA, alongside inhibition of fibroblast growth factor (FGF) signaling. These SANLPCs demonstrated typical pacemaker cell characteristics, including spontaneous action potentials, specific ion channel profiles, and responsiveness to neurohormonal signals. When transplanted into the apices of rat hearts, SANLPCs successfully integrated with the host tissue, initiating ectopic pacemaker activity, particularly under conditions of induced atrioventricular block [14].

Similarly, Chauveau et al. investigated hiPSC-derived embryoid bodies (EBs) in a canine model of atrioventricular block. The iPSC-CMs integrated into the heart tissue and exhibited pacemaker function. By the first week post-transplantation, the grafted cells demonstrated pacemaker activity, with 60–80% of the heartbeats originating from the injection site by the fourth week. The infusion of epinephrine further enhanced the rate of matching beats, indicating responsiveness to autonomic regulation. The study spanned up to 13 weeks, during which the iPSC-CMs maintained pacemaker function, though challenges such as variable rhythms and the need for immunosuppression persisted [15]. These limitations may be attributed to the use of an older differentiation method; thus, while the results are promising, the study emphasizes the necessity for refined differentiation protocols and enhanced integration strategies to achieve reliable outcomes.

In another study, Yu-Feng Hu et al. induced pacemaker cells from quiescent mature cardiac cells by overexpressing the vascular endothelial cell-adhesion glycoprotein cadherin. These transdifferentiated cells displayed both functional and morphological characteristics similar to those of native SAN cardiomyocytes in vitro. Furthermore, when injected into the left ventricles of rats, these SAN-like cells effectively functioned as sinoatrial nodes in situ [82].

Regarding hES-CMs, Izhak Kehat’s research involved transplanting hES-CM-derived embryoid bodies into the hearts of pigs. The results demonstrated that the transplanted cells survived, integrated with the host tissue, and paced the heart at rates similar to natural rhythms. Electroanatomical mapping confirmed that electrical activity originated from the graft site, indicating successful integration [83]. In another study, Tian Xu showed that hESC-CM EBs were capable of pacing the heart following cryoablation of the atrioventricular node in a guinea pig model, with optical mapping revealing that electrical signals spread from the site of transplantation to the surrounding myocardium. The transplanted cells sustained pacemaker function throughout the study period without causing arrhythmias or tumor formation [84].

## 6. Challenges in Clinical Translation

The success of these preclinical models demonstrates the promise of iPSC-derived biological pacemakers as a novel approach to treating bradycardia. However, it also highlights the challenges that must be addressed to ensure the safety, efficacy, and scalability of these therapies for clinical use. A key challenge for iPSC-derived pacemaker cells in clinical use is the risk of immune rejection [85]. Although deriving iPSCs from a patient’s own cells reduces this risk, genetic modifications or partially non-autologous cells can still trigger immune responses. Long-term immunosuppressive therapies can increase infection risk, highlighting the need for strategies such as gene editing for immune compatibility [86]. Ensuring the long-term functionality of iPSC-derived pacemaker cells is also crucial. While these cells can initially establish rhythmic activity, their performance may diminish due to factors such as cell loss, migration, or changes in behavior [87].

Variability in pacing rates due to differences in differentiation processes poses another challenge, which may require refined protocols and bioengineered scaffolds for better integration [64]. The potential risk of arrhythmias remains a significant concern, as the action potential of hiPSCs is influenced by factors such as cell density [88] and changes in culture protocols [89]. To enhance the efficacy of iPSC-based therapies, it can be beneficial to select fully differentiated cells through methods such as FACS to sort for SAN-specific markers (such as HCN or SHOX2); additionally, it can be beneficial to employ real-time monitoring with AI technology to oversee cell differentiation and detect variations among cell lines [90,91]. Scaling up the production of iPSC-derived pacemaker cells for clinical use is another hurdle, requiring efficient and reproducible differentiation protocols and adherence to strict manufacturing standards [92].

Developing a safe and effective delivery method is another essential aspect for the clinical application of biological pacemakers. Traditionally, delivery methods have involved highly invasive procedures, such as transarterial approaches or thoracotomy, which have limited their feasibility for clinical use [93]. To minimize invasiveness, some studies have employed intracardiac catheters through venous access to administer genetic therapy in large animal models [94]. However, these techniques can lead to partial retention of cells or genes, and the potential risk of systemic vector spread remains a concern after needle injection. Recently, a new device has been developed to enable direct intramyocardial transplantation of hiPSC-CM spheroids from large-scale cultures, achieving better distribution and retention within the myocardium compared to conventional needle injections [95]. This advancement may pave the way for clinical applications in human trials. Lastly, ethical and regulatory aspects must be taken into account, especially concerning the safety of genetically modified cells and the handling of patient data.

## 7. Future Directions and Research Opportunities

As the development of iPSC-derived biological pacemakers advances, several promising research directions have emerged. A hardware-free biological pacemaker offers a valuable temporary solution for patients needing interim pacing, particularly beneficial for high-risk individuals, such as the elderly or those with acute conditions that heighten procedural risk. This approach is especially advantageous for patients requiring pacemaker removal due to infections, acting as a temporary bridge until a new device can be implanted and thus reducing the risk of reinfection. Despite their potential, biological pacemakers have not yet demonstrated the capability to completely replace electronic devices in a fully hardware-free form. Achieving this level of functionality remains a crucial goal for future clinical applications as a viable standalone alternative to traditional electronic pacemakers.

## 8. Conclusions

iPSC-derived biological pacemakers offer a promising alternative to traditional electronic pacemakers for treating cardiac arrhythmias. These cell-based therapies can provide more natural rhythm control, reducing the need for maintenance and offering physiological responsiveness. Preclinical studies have shown their potential to integrate with host tissue and generate spontaneous pacing, but challenges remain, such as achieving long-term functionality, minimizing immune responses, and ensuring stability. Future research in gene-editing, tissue engineering, and differentiation methods will be key to optimizing these therapies for clinical use. With continued advancements, iPSC-derived pacemakers could revolutionize arrhythmia treatment, providing a patient-specific, durable solution that bridges the gap between current technologies and the body’s natural mechanisms.

## Figures and Tables

**Figure 1 cells-13-02045-f001:**
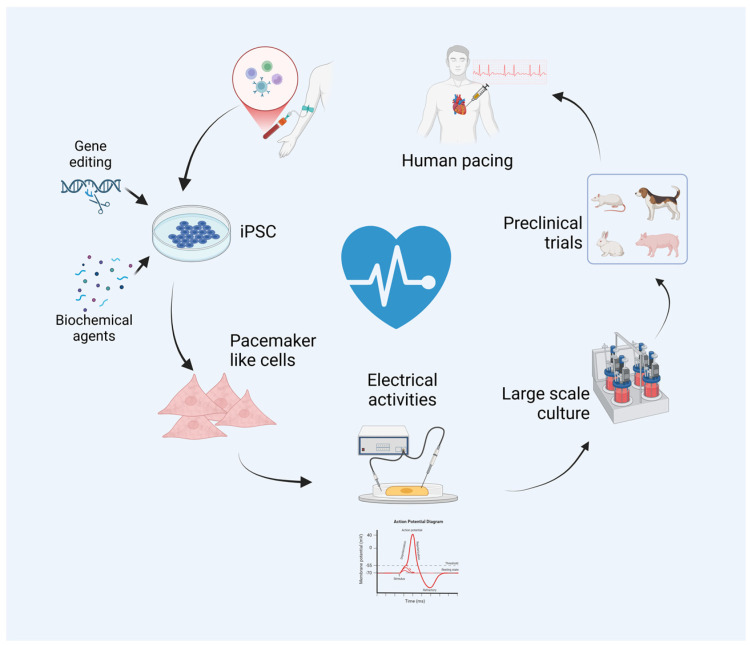
iPSC-derived biological pacemaker and its clinical application.

**Figure 2 cells-13-02045-f002:**
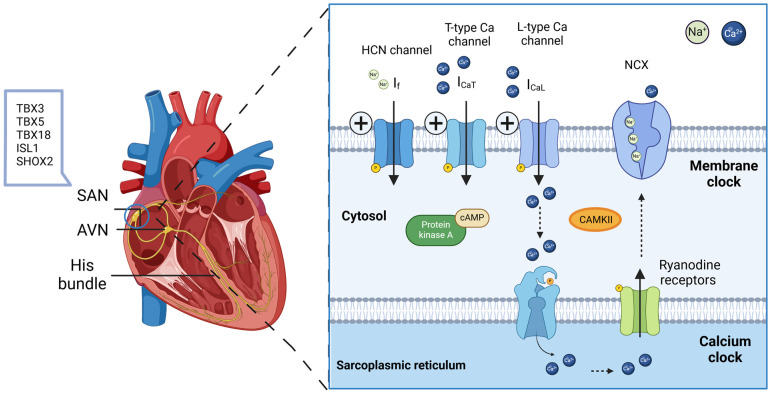
SAN coupled-clock system: A coupled-clock system consists of a membrane clock and a calcium clock. The membrane clock generates diastolic depolarization through pacemaker currents facilitated by HCN channels and L-type and T-type calcium channels. In parallel, the calcium clock operates in synchrony with the membrane clock, releasing calcium from the sarcoplasmic reticulum (SR) via ryanodine receptors, which activate the sodium–calcium exchanger (NCX) to assist with cell depolarization. Subsequently, calcium is reabsorbed into the SR by SERCA, regulated by phospholamban, to prepare for the next cycle. These clocks function together to create spontaneous depolarizations, and their activity is modulated by the autonomic nervous system through β-adrenergic and muscarinic signals that influence kinases such as PKA and CAMKII. (SR: sarcoplasmic reticulum; NCX: sodium–calcium exchanger; SERCA: sarcoplasmic Ca^2+^-ATPase; PKA: protein kinase A; CAMKII: calmodulin-stimulated protein kinase II).

**Figure 3 cells-13-02045-f003:**
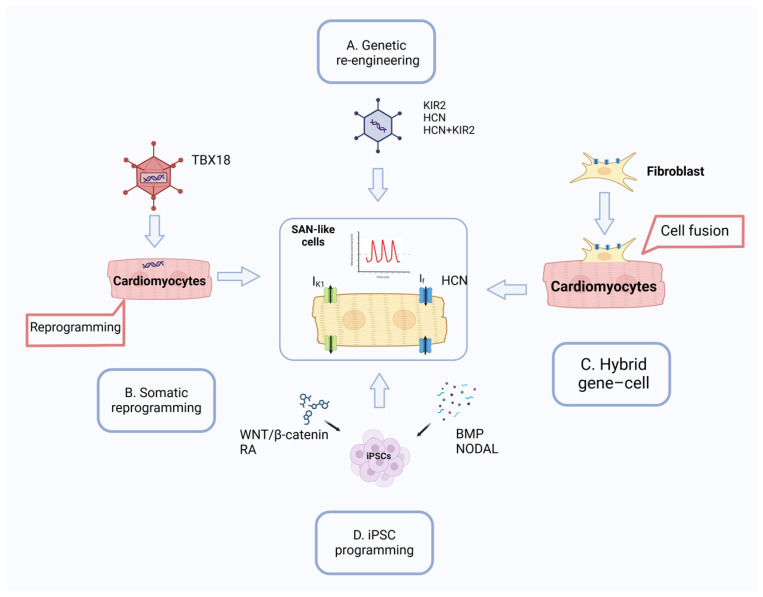
Strategy to develop pacemaker-like cells: (**A**) In the re-engineering strategy, virus vectors are utilized to enhance the expression of genes that encode ion channels within cardiomyocytes to induce automaticity. (**B**) Somatic reprogramming involves the overexpression of the T-box transcription factor TBX18 using virus vectors, transforming adult cardiac chamber cardiomyocytes into induced SAN-like cells. (**C**) A hybrid method employs cells such as fibroblasts to introduce ion channel genes to generate cardiac automaticity. (**D**) hESCs or iPSCs are differentiated with pharmacological manipulation to create SAN-like cells and transplanted into specific heart regions to integrate with the surrounding myocardium and establish biological pacing.

**Table 1 cells-13-02045-t001:** Advantages and limitations of pacemaker-like cell development strategies.

Strategy	Advantages	Limitations
A. Genetic Re-Engineering	Induces spontaneous electrical activity in non-pacemaker cells by introducing specific pacemaker-related genes, such as HCN channels.	Potential for arrhythmic complications due to unintended electrical disturbances; challenges in achieving stable and physiologically appropriate heart rates.
B. Reprogramming of Ventricular Myocytes	Transcription factors such as TBX18 or TBX3 can convert ventricular myocytes into pacemaker-like cells, closely mimicking natural sinoatrial node cells; advances in delivery methods, such as adeno-associated viruses, have improved clinical applicability.	Potential immune responses to introduced factors or vectors; risk of incomplete reprogramming, leading to partially converted cells with unpredictable behavior.
C. Hybrid Approach	Engineered non-cardiomyocyte cells (e.g., human mesenchymal stem cells) can modulate local cardiac activity through interaction with native cardiomyocytes; utilizes cells with potential immune-privileged properties, potentially reducing immune rejection.	Requires time for engineered cells to establish functional connections with native cardiac tissue; may result in low basal heart rates, limiting effectiveness; concerns about unintended cell migration and differentiation into undesired cell types.
D. iPSC-Derived Pacemaker Cells	iPSCs can be directed to differentiate into pacemaker-like cardiomyocytes using various protocols, offering a renewable and patient-specific cell source; transgene-free approaches reduce risks associated with permanent genetic modifications.	Differentiation efficiency and purity of pacemaker-like cells may vary, affecting consistency; functional integration into existing cardiac tissue remains challenging, with potential issues in establishing proper electrical coupling; risk of tumorigenesis.

**Table 2 cells-13-02045-t002:** Preclinical studies using pacemaker-like cells.

Study	Cell Type	Key Characteristics	Animal Model	Injection Site	Outcome
Protze et al. [14]	SAN-like pacemaker cells (SANLPCs)	Spontaneous action potentials; specific ion channel profiles; neurohormonal responsiveness	Rat	Apex	Successful integration with host tissue; ectopic pacemaker activity under AV block conditions
Chauveau et al. [15]	hiPSC-derived embryoid bodies (EBs)	Pacemaker function; 60–80% beats from injection site by week 4; responsive to epinephrine	Canine	Epicardium	Maintained pacemaker function up to 13 weeks; enhanced beat rate with epinephrine
Yu-Feng Hu et al. [82]	Quiescent cardiac cells transdifferentiated	Functional and morphological similarity to native SAN cells	Rat	Left ventricle	Effective pacemaker function in situ
Izhak Kehat [83]	hES-CM-derived embryoid bodies	Integration with host tissue; natural pacing rates	Pig	Posterolateral wall	Successful integration; pacing similar to natural rhythms confirmed via electroanatomical mapping
Tian Xu [84]	hESC-CM-derived embryoid bodies	Sustained pacemaker function, electrical signal spread from transplant site	Guinea	Anterior epicardium	Effective pacing without arrhythmias or tumor formation

## Data Availability

No new data were created or analyzed in this study. Data sharing is not applicable to this article.

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
