# Peer review of "iPSC-Derived Biological Pacemaker—From Bench to Bedside"

_cells, 2024, doi:10.3390/cells13242045_

Round 1

Reviewer 1 Report

Comments and Suggestions for Authors

This comprehensive review paper excellently addresses the development of biological pacemakers. It thoroughly describes the challenges and advancements in this field, making it highly valuable for both cardiologists and bioengineers.

Author Response

Comment: This comprehensive review paper excellently addresses the development of biological pacemakers. It thoroughly describes the challenges and advancements in this field, making it highly valuable for both cardiologists and bioengineers.

Response: We sincerely appreciate your positive comments and feedback.

Reviewer 2 Report

Comments and Suggestions for Authors

This is a review focused on our current understanding of Pacemaker cells derived from human induced pluripotent stem cells. Because of its narrow focus, the authors are able to go into some detail about the specific studies. The authors have completed a thorough review of the field

The review is comprehensive and provides a good overview of the field. The text is well-written and clear.  The references are up-to-date and also provide an excellent overview of the literature.

Author Response

Comment: This is a review focused on our current understanding of Pacemaker cells derived from human induced pluripotent stem cells. Because of its narrow focus, the authors are able to go into some detail about the specific studies. The authors have completed a thorough review of the field

The review is comprehensive and provides a good overview of the field. The text is well-written and clear.  The references are up-to-date and also provide an excellent overview of the literature.

Response: We sincerely appreciate your positive comments and feedback.

Reviewer 3 Report

Comments and Suggestions for Authors

The only concern is about the paragraph 4. I was expecting the presence of a small table with pro and limitations for each differentiation approach would help much more than the Fig.3. Or the authors can think about implementing the Fig. 3 with limitations and pros for each approach.

Comments on the Quality of English Language

English can be improved, consider the option to ask to a native speaker to read the manuscript.

Author Response

We greatly appreciate the reviewer’s comments.

 Comment: The only concern is about the paragraph 4. I was expecting the presence of a small table with pro and limitations for each differentiation approach would help much more than the Fig.3. Or the authors can think about implementing the Fig. 3 with limitations and pros for each approach.

Response: Thank you for your comment. In accordance with the reviewer’s comment, we added "Table 1. Advantages and limitations of pacemaker-like cell development strategies".

Comment: English can be improved, consider the option to ask to a native speaker to read the manuscript.

Response: Thank you for your comment. In accordance with the reviewer’s comment, our manuscript was edited in English by MDPI.